# The Role of Human Satellite III (1q12) Copy Number Variation in the Adaptive Response during Aging, Stress, and Pathology: A Pendulum Model

**DOI:** 10.3390/genes12101524

**Published:** 2021-09-28

**Authors:** Lev N. Porokhovnik, Natalia N. Veiko, Elizaveta S. Ershova, Svetlana V. Kostyuk

**Affiliations:** Laboratory of Molecular Biology, Research Centre for Medical Genetics, 115478 Moscow, Russia; satelit32006@yandex.ru (N.N.V.); es-ershova@rambler.ru (E.S.E.); svet-vk@yandex.ru (S.V.K.)

**Keywords:** repetitive sequences, satellite III, 1q12, copy number variation, copy gain, non-coding RNA, stress response, aging, genotoxic stress, pendulum model

## Abstract

The pericentric satellite III (SatIII or Sat3) and II tandem repeats recently appeared to be transcribed under stress conditions, and the transcripts were shown to play an essential role in the universal stress response. In this paper, we review the role of human-specific SatIII copy number variation (CNV) in normal stress response, aging and pathology, with a focus on 1q12 loci. We postulate a close link between transcription of SatII/III repeats and their CNV. The accrued body of data suggests a hypothetical universal mechanism, which provides for SatIII copy gain during the stress response, alongside with another, more hypothetical reverse mechanism that might reduce the mean SatIII copy number, likely via the selection of cells with excessively large 1q12 loci. Both mechanisms, working alternatively like swings of the pendulum, may ensure the balance of SatIII copy numbers and optimum stress resistance. This model is verified on the most recent data on SatIII CNV in pathology and therapy, aging, senescence and response to genotoxic stress in vitro.

## 1. Introduction

All eukaryotic genomes are highly repetitive. The estimated total fraction of repetitive sequences in human genome varies from approximately one-half [1] to over two-thirds [2]. While first studies on the human genome size in the 1990s elicited an estimate that the human genome harbored 50,000–100,000 protein-coding genes [3], by 2008, RNA-seq further identified an ocean of non-coding transcribed sequences, divided into long non-coding RNAs (lncRNAs), antisense RNA and miscellaneous RNA, and finally, in 2018, ~20,000 protein-coding genes, ~15,000 pseudogenes and, impressively, ~17,000–25,500 non-coding RNAs were catalogued [4]. Indeed, the coding RNA (mRNAs) covers merely 1–2% of the genome sequences, whereas a much larger part of the genomes is known to be actively transcribed to produce non-coding RNAs (ncRNAs) [5,6,7].

The scientific society converges that the terms “junk DNA” [8] or “selfish DNA” [9] relating to the repetitive sites of eukaryotic genomes should now be considered a carryover of the bygone era of understanding a gene “function” solely as coding for a messenger, transfer or ribosomal RNA. The previous “dark matter” of genome [10] has been found to be of great importance for the cell’s life, being involved in genome repair and integrity [11,12], disease development [1,13,14], gene regulation [15] and anti-stress protection (see Section 2 below). 

Historically, DNA fractions containing repeats were isolated by high-speed density gradient centrifugation of bulk DNA. Due to their different AT/GC content compared to the bulk DNA, repetitive DNA formed additional bands, giving rise to the term “satellite DNA” (satDNA) [16,17].

Human satellite DNA [18] is an essential part of so-called “constitutive” heterochromatin (defined as a transcriptionally inactive chromatin that remains compact throughout the cell cycle) comprising 3% of the genomic repetitive elements [19] and predominantly located at the centromeric and pericentromeric regions of the chromosomes, and their terminal ends—telomeric and subtelomeric regions [20,21]. In mammals, two types of satellite DNA sequences are usually described: major satellite repeats that are generally located at the pericentromeric regions and minor satellite repeats at the centromeric regions [22]. The pericentromeric “constitutive” heterochromatin quite recently appeared to be transcribed, and the transcription of SatIII non-coding RNAs is thought to play a pivotal role in a general stress response in human cells [23], as discussed in detail in Section 2. Furthermore, we review the most recent data on the role of SatIII copy number variation in modulating the SatII/III ncRNA-mediated stress response.

The experimental facts listed below are obtained on different satellite DNA classes located in different genome regions. It remains an open question to what extent the data obtained for different satellite DNA classes (II, III) and locations (1q12, 9q12) are relevant to the other classes and locations. When generalizing the empirical information and constructing a model, we postulated that the underpinning facts are universal.

## 2. SatII/III Transcription and Nuclear Stress Body Formation as a Universal Stress Reaction

There are several approaches to the classification of repetitive sequences in human genomes. They can be based on function [24], size [25] or a combination of them [1]. Satellite repeats can be further divided in microsatellites, minisatellites and macrosatellites [26,27,28,29,30,31]. Megasatellites are distinguished by some researchers, as well [32].

“Constitutive” heterochromatin in the pericentromeric areas has turned out to be just a historical term. The transcription of satellite DNA has been reported in different biological contexts [11,33,34], including development [22,35,36,37], cell cycle [22,34] and cellular response to stress [33,37,38,39], the latter being a general adaptive reaction in human cells [23]. An amazing link to the stress reaction is well established for the satellite DNA transcription.

### 2.1. Generic Stress Response: What We Learnt Recently

The best characterized anti-stress mechanism, which is triggered by a variety of stressors, involves the transcriptional upregulation of a set of genes coding for molecular chaperones [40]. Although this cascade is activated not only by thermal stress [41,42,43,44], but also irradiation, azetidine, UV light, cadmium salt [45] and hyper-osmotic conditions [23], the target genes are called heat shock genes for historical reasons. Heat shock response is considered to be a ubiquitous stress-response mechanism to protect cells against a broad spectrum of stress-induced damage.

In vertebrates, heat shock genes are under the control of a family of transcription factors, the so-called heat shock factors, HSF1 to HSF4 [46,47], each mediating the response to distinct forms of cellular stress (reviewed in [48]) or modulating HSF1 action. Among them, only HSF1 is critical for the activation of heat shock genes after classic stress impacts. HSF1 accumulates in an inactive form in the cytoplasm. Activation upon heat shock requires a number of events that ultimately results in HSF1 trimerization and transport to the nucleus, where the trimers bind to the heat shock elements (HSE) within the promoters of heat inducible genes [49].

One of the most striking effects of stress is the rapid and reversible redistribution of HSF1 into a few nuclear structures termed nuclear stress granules which form primarily on the 9q12 locus and chromosome Y [50] and secondary on several other chromosomes (including chromosome 1) enriched in SatII and SatIII sequences in humans. Within these structures, HSF1 binds to satellite II and III repeated elements at several pericentromeric regions in heat-shocked cells [51] and drives the transcription of these sequences into stable RNAs which remain associated with the source DNA locus for a certain time after synthesis. Other proteins, particularly splicing factors, were also shown to relocalize to the granules upon stress [52] of different nature. As a result, special structures termed “nuclear stress bodies” form. 

### 2.2. Nuclear Stress Bodies (nSBs)

The nuclear stress bodies (or nSBs) were discovered in the late 1980s and very soon after were associated with cellular response to stress factors [53,54]. They are transient subnuclear organelles clearly distinct from other nuclear bodies [55]. Intriguingly, these bodies have been described only in ape and human cells and never observed in rodents (reviewed in [56]).

The nuclear stress bodies assemble on blocks of satellite III DNA [57], inducing transcriptional activation of pericentromeric heterochromatin. It was observed that HSF1 binding, through the co-recruitment of the histone acetyl transferase CREB binding protein (CBP), initiated a series of events involving chromatin remodeling, the recruitment of RNA pol II, but not of pol I or III, and culminated with the production of SatIII transcripts [38,52,58]. Pol II is known to transcribe unique genes to form messenger RNAs. So, it seems at first sight unexpected that this type of RNA polymerase drives transcription of tandem repeats. However, SatIII sequences appeared late in evolution [59]. They are specific to the *Hominoidea* superfamily. Interestingly, structures comparable to nSBs were found in *Drosophila* and called omega speckles [60]. Similarly to SatIII RNA, omega speckles form in heat-shocked cells, but omega transcripts derive from a single copy gene, where pol II involvement is natural. We could hypothesize that hominid ancestors had had similar anti-stress molecular machinery based on a single gene sequence, but after a certain time in the past, evolutionary events have resulted in the replacement of that gene for a novel satellite sequence.

However, the adaptive stress response includes not only the activation of a series of protective genes, but also the temporal silencing of a wide variety of other genes. This process requires non-coding RNAs transcribed from satellite III repeats. The intriguing underpinned mechanism, though still partially elusive, is generally outlined at present, and HSF1 is a central player here, as well.

The mechanisms for local nuclear stress bodies having an effect on the global landscape of nuclear genome expression remain hypothetical in general. The global stress response, in addition to HSF inducible gene transcriptional stimulation, which is beyond this review, is composed of global transcriptional repression, modulating splicing activities, and HSF-independent transcriptional de-repression of genes located in the vicinity of nuclear stress bodies.

### 2.3. Global Effect of Nuclear Stress Bodies on Genome-wide Gene Expression: Molecular Traps

The activation of SatIII arrays participates in global stress-induced, genome-wide down-regulation of genome expression through transient sequestration of transcription factors [57,61].

Nuclear stress bodies may be viewed as transcription factories comprising a natural amplification of RNA pol II promoters. Although the number of transcription units is not yet defined, both the extent of the SatIII arrays and the size of HSF1 foci suggest that thousands of transcriptional units may be simultaneously activated [57]. Thus, the massive concentration of factors involved in the activation of SatIII sequences such as RNA pol II or the histone acetyl transferase CREBBP (also denoted as CBP) may result in the transient depletion of transcription factors from the surrounding nucleoplasm. This is consistent with the observation that, in heat-shocked cells, the formation of nSBs is followed by a global deacetylation of chromatin in the rest of the nucleus [62].

It was directly demonstrated that the recruitment of CREBBP to nSBs is SatIII-dependent, and that the loss of SatIII transcripts relieves the heat shock-induced transcriptional repression of at least a few target genes. Conversely, forced expression of SatIII transcripts resulted in the formation of nSBs and transcriptional repression even without a heat shock [61].

Thus, SatIII transcripts were shown to serve as a “sink”, as originally proposed [56], to recruit critical factors of the transcriptional machinery and thus aid in heat shock-induced gene silencing—at least for the target genes studied [61]. We discussed above the evolutionary conservatism between fly omega transcripts and human SatIII RNAs, both requiring RNA pol II. It seems reasonable to speculate that the resultant exhausting of the pool of RNA polymerase II is essential for the transitory down-regulation of protein translation, whereas involvement of the other types of RNA polymerase (I or III) in SatIII transcription could not produce the same effect.

The gene expression in heat-shocked human cells is also affected by transient targeting of specific splicing factors to SatIII RNAs, such as hSF2 or SRSF1 (serine/arginine rich splicing factor 1), other SRSFs and KHDRBS1 (also known as Sam68) [52]. The splicing reaction is either blocked or delayed by heat shock (reviewed in [63]). A similar hypothesis of intracellular depletion can be proposed for those factors involved in pre-mRNA processing. SatIII RNAs are stable components of nSBs and mediate the recruitment of a number of proteins involved in pre-mRNA maturation [52,64]. The recruitment of the splicing factor SRSF1 is mediated by the RRM2 (RNA recognition motif ) domain, which is critical for its activity in alternative splicing [52,65], whereas other RNA binding proteins, such as hnRNP HAP, are recruited through protein–protein interactions [64].

Splicing profiles are controlled by the relative abundance of antagonistic hnRNP and SR proteins. It is plausible that SatIII RNAs, by sequestering specific RNA binding proteins into nSBs, may shift splicing decisions, for instance, toward the synthesis of molecules involved in the cell defense to stress [57].

Alternative splicing affects more than 90% of cellular transcripts. The SRSFs recruited to nSBs are rapidly de-phosphorylated upon thermal stress exposure. The phosphorylation states of SRSFs are known to affect splicing patterns. During stress recovery, CDC-like kinase 1 (CLK1) is recruited to nSBs and accelerates the re-phosphorylation of SRSF9, thereby promoting target intron retention. A transcriptomic analysis revealed that the depletion of HSatIII lncRNAs, resulting in the elimination of nSBs, promoted splicing of 533 retained introns during thermal stress recovery. Thus, SatIII-dependent nSBs serve as a conditional platform for phosphorylation of SRSFs by CLK1 to promote the rapid adaptation of gene expression through intron retention (IR) following thermal stress exposure [66]. Among the five basic modes of alternative splicing (exon skipping, mutually exclusive exons, alternative donor site, alternative acceptor site and intron retention), IR is the rarest mode in mammals and, hence, is less studied. The anti-stress effects of IR are still obscure and require further studies.

### 2.4. Local Action of Nuclear Stress Bodies

Activation of SatIII facilitates transcription of nearby genes through *cis*-acting effects or by creating nuclear domains in which genes escaping heat-induced transcriptional repression would relocate. Transcriptional de-repression of genes adjacent to nSBs through position effects presumably starts from the loss of epigenetic repressive marks at pericentromeric loci following heat shock, which could abolish the transcriptional repression exerted by pericentromeric heterochromatin on the activity of the promoters of gene present in the *cis*- or possibly the *trans*- position through chromatin opening and recruitment of transcription factors [57].

Several examples have been reported in the literature in which gene inactivation is indeed associated with repositioning of repressed genes in the vicinity of these large blocks of heterochromatin. Based on these observations, a model has been proposed in which the repositioning of repressive genes in the vicinity of heterochromatin would be necessary for the maintenance of their repressed status through a position effect mechanism (reviewed in [67,68,69]). It is conceivable that transcriptional activation of SatIII sequences may impact the activity of other genes associated either on the same chromosome or in the nuclear space. This could be one of the multiple ways by which stress may induce a transient reprogramming of gene expression profiles.

## 3. SatII/III Copy Gain in Stress, Senescence and Cancer

Due to their highly repetitive nature, tandem repeats are known as a principal source of genome instability [32]. Human satellite DNA is characterized by increased instability with a pronounced quantitative polymorphism. The number of repeat units varies by several times at individual arrays, making tandem satellite DNA a prominent source of copy number variation (CNV) [12,28,30,31,70].

Open chromatin conformation necessary for transcription strongly assists in changing the copy numbers of satellite DNA due to unequal exchanges. Although the stress-induced satellite III DNA transcription with the formation of nSBs occurs primarily on chromosomes 9 [71] and Y [50], chromosome 1 satellite III DNA has been also shown to be decondensed, demethylated and transcribed in senescent cells and in A431 epithelial carcinoma cells [72].

Generally, senescence [73] and, especially, malignancies are well-known conditions to mimic permanent stress with the constitutively activated stress response. Cancer cells are chronically exposed to various stresses, such as lack of oxygen and nutrients, immune responses, dysregulated metabolism and genomic instability [74,75]. Moreover, many oncogenes are involved in stress response and cell survival, thus interfering anti-cancer therapy [76,77]. It seems reasonable, therefore, to expect an upregulated SatIII RNA-based stress reaction in cancer and/or senescent cells. Sure enough, aberrant overexpression of satellite repeats was found in pancreatic, cervix, colon and other epithelial cancers [35,78].

The human satellite II (HSatII) is the most differentially expressed satellite subfamily in epithelial cancers [78]. It constitutes the main component of pericentromeric heterochromatin on chromosomes 2, 7, 10, 16 and 22, and it is also found at chromosome band 1q12, where it is collocated with satellite III sequences [79]. Recently, an unanticipated mechanism of SatII copy number gain was uncovered in tumors. Physiological induction of endogenous human SatII RNA, as well as the introduction of synthetic HSatII transcripts led to these repeated transcripts being reverse transcribed into cDNA intermediates in the form of DNA/RNA hybrids. Single molecule sequencing of tumor xenografts showed that these HSatII RNA-derived DNA (rdDNA) molecules had been stably incorporated within pericentromeric loci, thus astonishingly incrementing the HsatII copy number. Suppression of reverse transcriptase activity using small molecule inhibitors reduced HSatII copy gain. Notably, the cancer-enriched human alpha satellite (ALR/Alpha) and simple satellite repeat (CATTC)n RNAs were also induced at high levels in xenografts with conjugate DNA copy number gains, suggesting a common mechanism of reverse transcription-mediated genomic expansion of particular repeat classes. Further analysis of whole-genome sequencing data revealed that HSatII copy number gain is a common feature in primary human colon and kidney tumors [80].

It still remains obscure to what extent these conclusions of non-canonical repeat transcription accompanied by reverse transcription-based copy gain could be extrapolated for SatIII and other types of moderate tandem repeats in different cancer and non-cancer conditions, including prolonged stress, senescence and pathology. A limited and undigested array of facts obtained in human and animal studies is available.

Somatic copy number alterations were analyzed using comparative genomic hybridization (CGH) in 75 cases of hepatocellular carcinoma in China. The most common alterations involved gains of 1q and 8q and a loss of 16q (50%), losses of 4q and 17p and a gain of 5p (40%) and losses of 8p and 13q (30%) [81]. These findings corroborate an early study of 36 histologically confirmed samples of hepatocellular carcinoma, of which 1q copy gain with a 1q12 breakpoint was detected in 23 cases. In parallel, DNA methylation in 1q12 was analyzed using Southern blotting with methyl-sensitive enzyme digestion. Interestingly, a strong correlation (*P* < 0.001) was found between the 1q copy gain with a 1q12 breakpoint and hypomethylated state of SatII sequences [82]. Obviously, such hypomethylation alters the interaction between the satellite DNA and chromatin proteins, resulting in heterochromatin decondensation and facilitating the satellite II/III transcription from 1q12 loci followed by the copy number gain.

Samples from 33 gastric cancer patients were collected in Saudi Arabia and analyzed using CGH compared to 15 normal gastric samples from the same population. The study revealed frequently copy number gains within several chromosomal regions including 1q12 among the most frequent [83]. Other findings of 1q overrepresentation in various types of cancer were obtained by increasing in situ hybridized site numbers in lymphoma and myeloma [84] and hepatitis B virus-related hepatocellular carcinoma [85].

Indirect evidence in favor of stress-based pericentric satellite repeat copy gain was acquired in animal studies on rodents. Although satellite III are primate-specific tandem repeats, there are similar pericentric repeats of MSat-160 in arvicoline rodents. Expansion of rDNA and MSat-160 pericentromere satellite repeats was revealed in the genomes of bank voles *Myodes glareolus* exposed to environmental radionuclides within the Chernobyl Exclusion Zone compared to animal subjects from adjacent non-contaminated areas. Notably, 18S rDNA and Msat-160 copy numbers were positively correlated in the genomes of bank voles from uncontaminated areas, but not in the genomes of animals inhabiting contaminated areas [86].

Cellular senescence was first described by Hayflick and Moorhead in the 1960s [87] as the irreversible arrest of cell growth following prolonged cultivation. Telomere shortening is deemed a marker and the key mechanism that underpins the process of replicative senescence [88]. Later, stress was shown to play a major role in the induction of premature senescence in vitro [89].

Cellular senescence results from a variety of stresses [90]. Seeming at first sight controversial, the degenerative and hyperplastic pathologies of aging are at least partly linked by a common biological phenomenon: a cellular stress response known as cellular senescence [91]. Besides environmental stress, some other physiological and pathological conditions are known to lead to the activation of pericentric sequences in human cells, suggesting the expectation of SatIII expression as a generic stress reaction during natural or induced senescence. Indeed, the expression of pericentric transcripts has been shown to occur during replicative senescence at late passages of both primary fibroblasts and cancer cells [72,92]. Higher-order unfolding of satellite heterochromatin is a consistent and early event in cell senescence. Large-scale distension or “unraveling” of peri/centromeric satellites occurs in all examined models of human and murine senescence [72,93]. Large-scale distension of satellites registered as an elevated area of the hybridization signal in the interphase nuclei may be associated with satellite copy number gain rather than chromatin decondensation.

An accelerated aging model also corroborates the existence of SatIII transcription in aging/senescence. In fibroblasts derived from patients with Hutchinson–Gilford progeria syndrome, a complete loss of the heterochromatic marks was followed by the expression of chromosome 9 specific SatIII sequences [94].

A progressive expansion of satellite III (1q12) copies was observed in replicative senescence and during natural aging [95] (for details, see Section 5.1 below).

Summarizing, we propose a universal mechanism to launch a process of augmentation of SatIII (1q12) and other pericentric tandem repeat copy numbers in chronic stress (Chernobyl zone), replicative senescence, normal and accelerated aging and various cancers. We hypothesize that in all these conditions, the satellite repeat copy gain is associated with stress-induced transcription of non-coding RNAs from the satellite DNA, with subsequent reverse transcription and cDNA reintegration. 

## 4. Too Much of a Good Thing: Vulnerability of Cells with Large SatIII (1q12) Blocks

What can be the possible consequences of SatII/III copy gain for the cell? As we can imagine, an increment of the satellite units obviously results in elevated transcription yielding more SatIII RNAs and thus strengthening the stress response. Hence, the cell line presumably becomes more resistant to the forthcoming stresses.

For tumor cells, higher stress resistance means declined response to anti-cancer therapy and worse prognosis. Survival data of 46 samples of colorectal cancers were analyzed to compare tumors with SatII gain vs. no gain. Kaplan–Meier tests showed a significant reduction in overall survival in the SatII gain cases, with a median overall survival of 1096 vs. 1881 days; log-rank *p*-value = 0.034 [80]. The patients with satellite copy gain in the tumor apparently have bad prognosis because of more resistant malignant cells compared to those with the baseline repeat count.

The animal model described above also corroborates the hypothetically increased viability of organisms carrying increased numbers of satellite copies in their genomes. In the conditions of heightened radiation background, bank vole genomes with increased Msat-160 copy numbers have received a selective advantage in the successive generations [86].

The location of satellite III stretches in chromosome Y described above [50] may reflect higher stress resistance, evolutionarily required for males.

However, experimental facts began to appear that the increasing abundance of pericentric satellites in the cell pool is a process lasting to a certain limit rather than ad infinitum, and possibly, reversible. Thus, in a large sample (*n* = 840) of patients with schizophrenia, a disease in which elevated reactive oxygen species levels and declined antioxidant statuses in the brain and peripheral tissues was reported [96,97,98,99,100,101,102,103], and the genomes of patients were surprisingly found to contain significantly fewer copies of SatIII than mentally healthy individuals (*P* << 10^−17^) [104].

There is nothing surprising about that. The stress reaction drastically changes the transcriptional landscape on a whole-genome scale. Expression of heat shock proteins is boosted, whereas most other genes undergo global silencing and aberrant splicing. Substantially, all the housekeeping and specialized processes are aborted or changed because of depletion of the transcription machinery parts. It helps to save resources, protect the chromosomes against damage and reduce the vulnerability of cellular machinery until the stress impact has finished. However, this response is transient by its very nature. This is just the lesser of two evils, and the longer it lasts, the more deleterious it becomes.

A low level of SatIII RNAs is detectable even in unstressed cell with an approximately 10,000-fold induction from the baseline after thermal shock or another stress treatment such as free radicals, heavy metals, ultraviolet or hyper-osmotic stress [23]. In case of a very long SatIII tandem array, the elevated baseline transcription can be anticipated, entailing partial depletion of polII, CREBBP and other factors even in unstressed conditions with a negative effect on the vital cellular processes.

In a series of experiments when HeLa cells were transiently transfected with antisense oligos for the knockdown of SatIII transcript, exposed to a heat shock (42 °C for 1 h), and cell viability was measured after heat shock release, the blockage of SatIII transcription led to a significant reduction (~70%) in cell survival as compared to cells that were transiently transfected with the control oligos and exposed to a heat shock. However, the SatIII antisense oligos did not affect the survival of cells that were not exposed to the heat shock, suggesting that the SatIII transcripts are required for survival only under thermal stress condition. The authors next wanted to estimate whether forced expression of SatIII transcripts would alter the viability of cells that are not exposed to a heat shock. Transient expression of a SatIII mammalian expression construct (pcDNA-SatIII) resulted in a paradoxically lower survival rate as compared to cells transfected with an empty vector or the bacterial expression vector SatIII repeats. Intriguingly, cells that expressed the cloned SatIII repeats exhibited significantly lower survival rates when exposed to heat shock, suggesting a toxic effect of Sat3 transcripts when ectopically expressed [61]. These experiments support our suggestion of the “lesser evil” of anti-stress response, especially when it is aberrantly long-lasting and/or induced ectopically.

Direct findings that evidence vulnerability of excessively long SatIII arrays were obtained in cytogenetic studies focused on human 1q12 blocks and their nuclear repositioning in health and pathology.

It has long been known that within the interphase nucleus, human chromosomes are spatially organized as chromosome territories (CT) or domains [105,106,107,108]. One of the best studied aspects of chromatin radiality is how individual CTs or selected gene loci are arranged with respect to the nuclear lamina (reviewed in [109]). Larger chromosomes are generally more peripherally located compared to smaller ones [110,111]. The radial position of CTs with respect to the lamina is associated with the size of the chromosomes in base-pairs, but also with the density of genes along each chromosome [112,113,114,115,116,117,118,119,120,121,122,123]. According to the “genomic bodyguard” hypothesis [124], constitutive peripheral heterochromatin may protect euchromatin from chemical mutagens and X-ray radiation because its location is adjacent to the nuclear membrane during interphase. Thus, it is positioned to absorb more environmental hazards than central euchromatin and may shield the DNA in euchromatin from damage and mutation [125].

The CT position is not fixed rigidly, but changes in response to environmental impacts, such as ionizing radiation (IR), reactive oxygen species and other deleterious impacts, demonstrating regular spatial shifts [126,127]. During the development of the adaptive response, the pericentromeric loci of homologous chromosomes appear to move from the perimembrane sites of the cell nucleus and approach each other for a possible repair of double-stranded breaks of DNA in the process of homologous recombination. After an exposure to X-ray radiation at an adapting dose of 10 cGy, transposition of the chromosomal pericentromeric loci and the accompanying activation of the chromosomal nucleolus-forming regions were observed in the irradiated lymphocytes [128].

The 1q12 site harbors numerous satellite II/III arrays. SatIII homology with human genome database was analyzed and the repeat was found to be located on eight human chromosomes: 1, 2, 7, 9, 10, 16, 22 and Y. Fragments of the Y chromosome with low homology are rare. The remaining chromosomes do not contain fragments homologous to the SatIII probe. More than half of the 1000 homologous SatIII sequences found in the database are localized within the first human chromosome 1q12 region. The first chromosome fragments found in the database contain dozens of tandem repeats homologous to the SatIII. Fragments of the remaining chromosomes according to the database contain fewer number of the SatIII units [104]. 

A fragment of the pericentromeric satellites III was studied using DNA probe PUC1.77 [129] specifically hybridizing to the 1q12 pericentromeric heterochromatin of the first chromosome. In lymphocytes, endothelial cells and mesenchymal stem cells, these heterochromatin blocks localize close to the nuclear membrane. Under low doses of ionizing radiation, 1q12 sites move toward the nucleus center deep into the nucleus. It is considered as the initial stage of the adaptive response [130,131,132].

The adaptive response induced by low-dose ionizing radiation involves an alteration of the chromatin spatial configuration that is necessary to re-shape the expression profile of the genome in response to stress. The 1q12 heterochromatin loci movement from the periphery to the center of the nucleus is a marker of the chromatin configuration change. It was hypothesized that a large 1q12 domain could affect chromatin movement, thereby hindering the translocation and inhibiting the adaptive response (AR) [133].

The change in the SatIII (1q12) position in the nucleus under the stresses and the absence of this transposition in some cells were found in a number of studies [130,134,135,136,137,138]. Notably, the cells that, for one reason or another, had not changed the 1q12 localization in response to irradiation, died during the cultivation [130,135,137]. Thus, one can expect that the 1q12 locus sizes (i.e., SatIII content) may be important for the realization of the chromatin spatial configuration necessary for AR. The cells with redundantly large 1q12 loci possibly may not be able to chromatin rearrangement due to steric obstacles. Such cells are expected to die first in chronic stress conditions. In this case, the population should accumulate the cells with small 1q12 loci sizes, and a decrease in the SatIII content should be found in isolated DNA. This hypothesis was verified in studies of the response of human cultured lymphocytes and mesenchymal stem cells (MSC) to low doses of IR, PHA stimulation and hydrogen peroxide. The response to the stress and proliferative stimuli associated with the 1q12 loci movement toward the nucleus center was not realized in the cells with a too large 1q12 loci size [133].

The 1q12 regions are detected using Fluorescence In Situ Hybridization (FISH) as two fluorescent signals. In lymphocytes, the position of the FISH signal in the projection plane (circle) depends on how the nucleus is located on the slide during the sample preparation. Image processing includes determining the gravity center of the signal, the signal radius vector (*r1* and *r2*) value, the distance and angle between the signals (*d* and *α*), the signal area and the radius and the nucleus area. The radius vector *r* is normalized to the value of the nucleus radius and changes from 0 (nucleus center) to 1 (nucleus surface). Most signals are located in the area corresponding to *r*-values > 0.75; that is, in lymphocytes from healthy donors, 1q12 loci detected by the SatIII DNA probe are located near the nuclear envelope.

Low-dose IR or hydrogen peroxide (10 μM, 3 h) significantly changes the 1q12 loci position in the nucleus. In response to stress, 1q12 loci move from the perimembrane region (*r* > 0.75) deep into the nucleus and converge with each other. A similar 1q12 loci movement was also observed when a proliferative stimulus PHA had been applied [133].

Surprisingly, the stress-induced 1q12 loci movement depended on the locus size. In the control cells, the analysis of dependence of the FISH signal area on the radius vector *r* showed no differences between cells with *r* > 0.75 and *r* < 0.75. However, in activated lymphocytes, there were significant differences in signal areas characterized by different *r*-values. Signals with *r* < 0.75 occupied a much smaller area than signals with *r* > 0.75. Thus, irradiation and PHA stimulation of healthy donor lymphocytes induced translocation of 1q12 loci that occupy a relatively small volume, deep into the nucleus. Loci of large size remain adjacent to the nuclear membrane [133].

Interesting results were obtained measuring SatIII content in irradiation (50 cGy) exposed lymphocytes during longer cultivation (7 days) after irradiation. During the cultivation, some cells died and had signs of apoptosis and necrosis. The mean SatIII content was almost twice reduced as compared to the start of cultivation. At the same time, the population appeared to contain predominantly cells with small 1q12 size. Thus, in response to oxidative stress, the pool of cells underwent selection by SatIII content. Cells with low repeat abundance had an advantage. Since this repeat is distributed throughout the 1q12 site, it may be assumed that primarily cells with large 1q12 loci occupying a large nucleus volume died due to the stress. The same effect was then shown for MSC [133].

Five human skin fibroblast (HSF) lines demonstrated the SatIII increase in the genomes of older cells during replicative senescence, suggesting the copy gain of the satellite repeats. The fewer copies the HSF genome had harbored before the experiments started, the more numbers of passages this line underwent before division was arrested. The copy gain appeared to be inversely proportional to the baseline SatIII content. These findings also corroborate the assumptions of SatIII expansion and toxicity of large SatIII clusters [95].

Forces and molecular mechanisms that shape the radial configuration of the 1q12 loci under the stress action remain elusive, with prospective comprehension of the role of dynamic interactions of chromatin with the nuclear lamina-associated protein complexes [139,140,141,142], possible role of caspase-3 [137], phase separation process [143,144] and/or histone modifications that might also play a role in shaping chromatin configuration. The treatment of the cells with a histone deacetylase inhibitor resulted in the relocation of the chromatin loci from the nuclear periphery toward the center [145]. In addition, transcriptional activity on the whole-genome scale was recently suggested to be the main force that changes the radial chromatin configuration in the nucleus [146]. 

## 5. “Pendulum” Model

We presented above a series of facts suggesting that there may exist a widespread or even universal mechanism of stretching the mammalian pericentric satellite tandem repeat clusters in the conditions of frequent or permanent cellular stresses. The mechanism was directly demonstrated in cancer cells, and indirect evidence was obtained in human aging and animal population studies. Extrapolating, the satellite cluster shall sooner or later become excessively long.

As scrutinized by means of an example of human 1q12 loci, too long satellite arrays make the cell stress-intolerant, dying after further stress impacts. It results in positive selection for cells with short satellite arrays, which can elongate during the next stress impacts, and so the cycle repeats.

Thus, we propose here a model of alternating growth of satellite DNA arrays and sweep them out by negative cell selection or via another mechanism yet to be elucidated. By analogy with swings of the pendulum, we call it the “pendulum model” (Figure 1).

Further studies are required to address the natural question regarding the origin of the minor fraction of stress-resistant cells with short satellite clusters existing or appearing among the cells with long clusters lacking stress resistance and perished by stressors. The existence of such fraction is proven by the high variance of the satellite DNA content observed in samples with an elevated average abundance of satellite DNA. The first speculative explanation is that, by an unknown cause, a subset of cell avoids the satellite expansion, serving as a reserve stock for future. Alternatively, a mechanism of excision, unequal exchange or another avenue to remove the excessive satellite repeats acts to produce, at low frequency, cells with shortened satellite DNA clusters in every cell division.

Below is an attempt to verify the pendulum model by applying it to some published experimental data obtained in various conditions.

### 5.1. Aging

In a sample of 557 subjects aged 2 to 91 years, a trend of increasing mean SatIII content with age can be clearly seen (Table 1). A difference between the Children and Senile groups and the Adult 1–3 groups is observed [95].

Growing SatIII content with age is coherent with the idea of satellite repeat expansion during habitual stresses throughout the lifetime. At the same time, conspicuous is the fact that the variance is also increasing with age, as SD gradually grows from 2.7 in Children to 8.5 in the Senile group. Moreover, the minimum value in the Adult 1–3 and Senile groups is even less than that in Children, suggesting that the low-copy cells are produced from high-copy parental cells rather than retained through a series of cell divisions in a minor fraction of subjects who have avoided strong stresses during the lifetime [95].

### 5.2. Stress Impacts

Five human skin fibroblast (HSF) lines were exposed to potassium Cr(VI) salt, which is known to induce oxidative stress with severity depending on the Cr(VI) concentration [147,148,149,150,151]. Moderate stress at 4 μM of Cr(VI) induced SatIII copy gain in the DNA of early passage cells with low repeat amount (HSF-66 and HSF-61) to about the same extent as replication senescence. Strong stress at 6 μM of Cr(VI) reduced the observed effect on these cells. Strains with a high baseline SatIII content (HSF-57, HSF-49 and HSF-41) reduced SatIII abundance in their genomes in response to genotoxic Cr(VI) action. Therefore, in response to genotoxic stress, cultured HSF with low SatIII content incremented their SatIII content, whereas HSF with high SatIII content otherwise decreased the abundance of SatIII repeats, thus demonstrating the oppositely directed "swings of pendulum" [95].

In subjects who had a chronic occupational exposure to low-dose ionizing radiation, the picture was similar to aging. While in the control cohort, SatIII content varied from 9 up to 33 pg/ng (mean value of 21 ± 5 pg/ng), in the gamma neutron-irradiated cohort, SatIII varied from 10 to 40 pg/ng (24 ± 6 pg/ng), and in the tritium-exposed cohort, it ranged from 1 to 176 pg/ng (29 ± 26 pg/ng). Again, as in the case of aging, growth of the mean value with stress was followed by elevated variance and wider range. Moreover, the minimum value in the tritium exposed cohort was less than the least value in controls, suggesting that the “reverse swing of the pendulum”—clearance of large 1q12 loci had occurred in some individuals [152].

### 5.3. Pathology (Schizophrenia)

Oxidative stress was found to account for at least part of cases of schizophrenia (see above, Section 3). However, this disease is multifactorial, with a complex of genetic and organic causes in the pathogenesis. In particular, fetal hypoxia can be a traumatic factor triggering the progression of schizophrenia during adulthood. In a large sample of schizophrenia patients, patients who had experienced fetal hypoxia were found to contain the same amounts of SatIII DNA in their genomes as healthy controls (*P* > 0.2), while patients without hypoxia in medical history, i.e., with genetic background of the pathogenesis, surprisingly contained far fewer copies than healthy controls (*P* << 10^−17^), as noted above. SatIII copy numbers in schizophrenia cases negatively correlated with the PNSS (Positive and Negative Syndrome Scale) score that shows disease severity. It is not clear whether low abundance of satellite DNA was a direct cause of more severe signs or merely an indicator of stronger stress the patient had experienced.

Interesting results were obtained studying the therapy effect on the SatIII content in leukocytes. Antipsychotic therapy is known to induce oxidative burst [153]. Patients with originally very low SatIII abundance (6–12 pg/ng DNA) demonstrated an increase in this repeat content in the course of antipsychotic therapy. Conversely, the SatIII content decreased after the course of therapy in patients with relatively high baseline SatIII content (25–35 pg/ng DNA). Comparison of these two samples showed that the treatment in general did not lead to change in the SatIII DNA content (D = –0.17, α = 0.11; *P* > 0.2). The SatIII content also did not depend on the type of neuroleptics used (*P* > 0.1) [104]. Once more, we can see the opposite effects (“swings of pendulum”) of stress resulting in SatIII copy gain to a certain limit and then a reduction in SatIII abundance during continuing stress.

## 6. Conclusions

We accumulated and reviewed an amount of data large enough to hypothesize that the transcription of pericentric 1q12 loci is involved in the standard stress response in human cells. We believe that a process of copy gain is a universal component of this response. However, the stress-induced SatII/III repeat expansion occurs to a certain limit. After the limit is reached, another mechanism is activated to reduce the number of SatIII copies. The mechanisms underlying the augmentation and removing excessive satellite repeats remain obscure. Nothing is known to us how the cell detects the limit to switch on the copy number reduction machinery. Perhaps, comparison with microsatellite expansion in unique gene promotors might help. However, to what extent the pericentric satellite stress-induced expansion is similar to oligonucleotide expansion resulting in some hereditary diseases (among them, fragile X chromosome is the best studied example) remains to be studied. The effects of stress-induced transcription from satellite DNA in 1q12 locus are an almost undiscovered field waiting for future studies.

## Figures and Tables

**Figure 1 genes-12-01524-f001:**
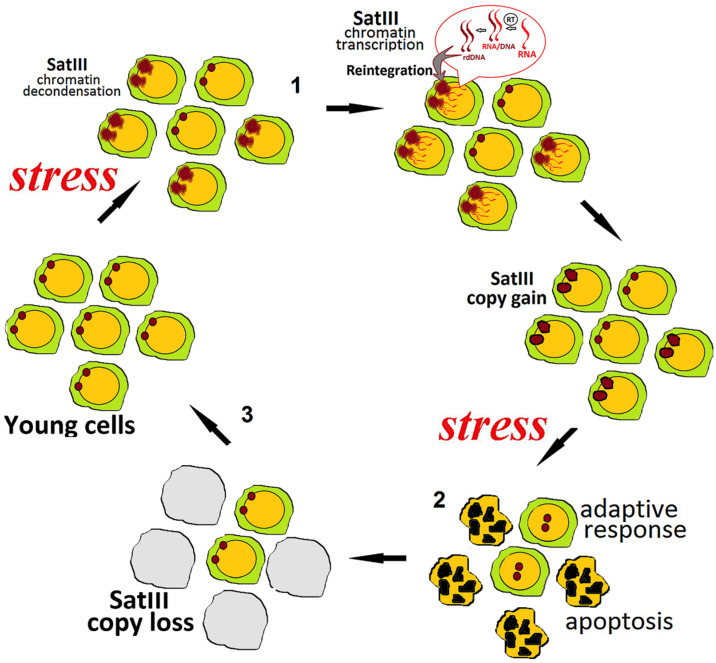
A “pendulum” model of the dynamics of SatIII copy number. (1) The mechanism of SatIII expansion in a fraction of cells under stress or aging; (2) cells with excessively large SatIII clusters perish during the repeated stress; (3) the low copy number (“reserve stock”) cells have survived and given rise to the future generations, which can then accrue their SatIII copy number and thus repeat the cycle.

**Table 1 genes-12-01524-t001:** SatIII content depending on the age (our previous data reported in [95]).

Group and Age Range, Years	Mean SatIII, pg/ng	Range, pg/ng	Standard Deviation (SD), pg/ng
Children, 2–12 years old	14.7	11.0–25.1	2.7
Adult 1, 17–36 years old	21.2	5.7–39.0	7.2
Adult 2, 37–56 years old	21.7	7.0–40.0	6.0
Adult 3, 57–76 years old	22.2	9.4–39.0	6.0
Senile, 77–91 years old	23.5	7.2–39.6	8.5

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
