# Peer review of "The Role of Human Satellite III (1q12) Copy Number Variation in the Adaptive Response during Aging, Stress, and Pathology: A Pendulum Model"

_genes, 2021, doi:10.3390/genes12101524_

Round 1
Reviewer 1 Report
In this review, the authors present the role of human satellite III DNA copy number in response to stress, aging, and pathology. They show that transcription of 1q12 loci is involved in the standard stress response in human cells. Also, they suggest an interesting model what they call the "pendulum model" that can explain dynamics in human satellite III DNA copy number. This work is relevant in the field of repetitive DNA.
Minor point:
1) Through the paper, you use different synonyms for satellite III DNA (SATIII, SatIII, Sat III, Sat3, satIII).
2) Check one more time for errors in spelling (line 173: premRNA; line 241: HSatIIcopy) and dots (line 75).
3) Lines 304-307 and 508-510 are similar. Try to formulate this satellite III DNA effect on aging in paragraph 6.1. Aging (line 507).
Author Response
Dear Reviewer,
We appreciate very much your valuable efforts in reading and estimating our manuscript. We are glad to receive a positive opinion from you and thankful for the estimable revisions you suggested.
Please find below our replies to each point of your review report:
1) We have unified all the different synonyms for satellite III DNA (SATIII, SatIII, Sat III, satIII) to the common abbreviation 'SatIII', except for Sat3, which has remained and, moreover, added to the Abstract ("SatIII, or Sat3"). The reason is that some authors completely avoid using Roman numerals. In order to find such reports, the reader has to know that search for 'Sat3' query is also necessary.
2) The spelling has been checked for errors through the whole text.
3) In lines 304-307, we left just a reference for paragraph 6.1 (line 507), where the satellite III DNA effect on aging is formulated.
Reviewer 2 Report
The manuscript by Porokhovnik et al. reviews functional importance of satellite repeats (focused mostly on satellite III at 1q12 locus) in human genome response on different stress conditions (including aging and pathological states), through satDNA transcription, copy number changes and intranuclear relocations of chromatin. A pendulum model of gain and loss in adjusting satellite repeat copy number is proposed. Information given is detailed and interesting, supported by extensive overview of the literature. It is within the scope of Genes and I recommend it but have some observations to be considered.
Abstract: Please consider starting the first sentence from “The pericentromeric satellite repeats recently…” for easier readability. The link between transcription of satellite repeats and copy number changes, which is the central issue of this review, should be more clearly stated. SatII is also often referred in the text but it is not mentioned in Abstract (or in the title). SatIII should appear after first mentioning, line 12.
Introduction:
Line 43: suggest rephrasing to “…repetitive DNA formed additional bands, giving rise to the term ‘satellite DNA’ (satDNA)”.
Lines 45-56: I suggest rewriting this and the following paragraph (58-75); they may be even merged. Characteristics of SatIII, SatII and alpha satDNA as main actors in this work should be briefly described (e.g. monomer length, approx. array length, formation of HOR, abundance, localization). Some recent estimation of satDNA abundance in human genome should be indicated, for example alpha satDNA is about 10%, located in the centromeric and pericentromeric regions of all chromosomes (for example, McNulty and Sullivan 2018; there are also comparable estimations from pre-genomic era). State briefly how are SatII and III positioned with regard to alpha. In the manuscript, alpha is mentioned only in line 243; as it can also be enriched in transcripts, would it be possible that at least part of it has similar dynamics as SatIII (for example, monomeric repeats in pericentromeres)?
SatII is invoked in several paragraphs, although the conclusion and the title refer only to SatIII, consider highlighting links between them (or differences)? The sentence in lines 49-51 probably refers to mouse major and minor satDNAs. Nevertheless, this information is not needed here and can be suppressed. Please also change RNA to DNA in Line 53. In addition, “non-coding” can be deleted.
Line 67, please rephrase (to avoid “called”): …arrays of head-to-tail tandem repeats.
Lines 71-75: The Table 1 in (32) is misinterpreted: minisatellite repeat units are not up to few hundred bp (line 73). In general, I don’t see too much relevance for the topic discussed here to present the subdivision which is not used further in the text, and can only confuse the reader. There are many recent reviews focused onto satDNAs, and maybe it would be easier just to cite some.
Figure 1: why is the block “structural components of chromosomes” shown separated from the “tandem repeats” block, as satellite DNAs are major components of centromeres and pericentromeres in human, and telomeres are also built of tandem repeats. Minisatellites and microsatellites are interspersed throughout the genome, as could also be monomers and short arrays of alpha satDNA (for example, Feliciello et al. 2020). Please rearrange this figure for easier understanding.
In the paragraph lines 99-108 are explained granules forming on chromosomes enriched in SatII and III (here in the text SAT is in capitals, elsewhere is Sat, satellite or HSat, please standardize) concluding that “As a result, special structures termed “nuclear stress bodies” form.” But SatII is not mentioned in line 115: “The nuclear stress bodies assemble on blocks of satellite III DNA” – please modify / clarify. Transcription is further explained for SatIII, but from the line 232 on also for SatII, where its role in the copy number gain is given. This section is named “SatIII copy gain in stress, senescence, and cancer” (line 212). Maybe SatII can be also added to this title? An extra sentence explaining that both SatII and III pronounce the same / similar effects under stress and that the first part of this section is focused on effects of SatIII could be added.
Please consider adding SatII also in the subtitle Line 314, because in the first sentence of this paragraph both are mentioned. In line 315 (and elsewhere) I suggest to replace the word “speculate” with propose or hypothesize. Lines 316-317, please explain if this is also valid for SatII.
Line 376: citation 109 is not a review, please check
Line 394: “The 1q12 site harbors numerous satellite II/III arrays.” Again, it seems that processes on satellites II and III go in parallel, could also SatII be mentioned in the title of this review?
The model in Figure 2 summarizes dynamics of SatIII – please indicate if it also applies to SatII. As it integrates data presented in this review, I think that more details can be given in the legend. Each step in the figure could be numbered and explained in the legend starting with young cells. I couldn’t find what is “low ROS”, please explain. Why the first stress is aging (in small font), I understood that different causes could be the trigger? Of what kind is stress (large font) before 2 (adaptive response)? Please explain. Why not chromatin relocation in the first step, just transcription and copy number gain in a fraction of cells? Is there transcription after 2nd stress (large font) or only relocation in adaptive response? What differentiates that there is no adaptive response in the first instance? Please clarify in the figure and fig. legend. “Minor fraction” (line 497) is not minor in the figure.
Table 1. The largest difference seems to be between the children group and the adult 1, not between the adult 3 and senile, at least it looks so to me. Please clarify. Concerning the range, in addition / instead of the mean, could it be expressed as the median value?
Conclusions. Line 569, please add if SatII copies are, or are not reduced in number. I suggest to add to this section also about adaptive response.
Author Response
Dear Reviewer,
We appreciate very much your valuable efforts in reading and estimating our manuscript. We are glad to receive a positive opinion from you and thankful for the estimable revisions you suggested.
Please find below our replies to each observation to be considered as specified in your review report:
1) the Abstract has been re-wrote to start from “The pericentromeric satellite repeats recently…” for easier readability. The link between transcription of satellite II/III repeats and copy number changes has been more clearly stated, with mentioning SatII in Abstract. The abbreviation SatIII now appears after first mentioning.
2) Introduction:
a) Line 43: we rephrased to “…additional bands" instead of "satellite bands".
b) We have decided to delete section 2 (lines 58-75) with Figure 1. As you truly noticed, there is little relevance for the topic discussed here to classify the satellite subdivision which is not used further in the text, and can only confuse the reader. There are many recent reviews focused onto satDNAs. So, we have just cited The references have been relocated to the beginning of former section 3 (now section 2). Inside our team, some co-authors have all along wanted to remove the satDNA classification section, and after had you kindly proposed so, this point of view won.
3) Main text
c) we have standardized SAT, Sat, sat to the unified variant Sat.
d) with regard to several places in the manuscript, such as lines 115, 212, 316-317, 394, former Fig. 2 (now Fig. 1), 569 you naturally asked if it is relevant to SatIII only or to SatII as well. Indeed, SatII, SatIII, and/or SatII/III are mentioned mixedly within the text. This is because different reports involved different regions, and it remains an open question to what extent the data obtained for one satellite DNA classe and location are relevant to the other classes and locations, for example, whether data obtained on chromosome 9, can be extrapolated to chromosome 1, and vice versa, and so on. We have now stated it directly in the last paragraph of Introduction. As the manuscript is a review, it contains a lot of citations. We everywhere tried to mention exactly the same satellite type/location as in the report being cited. But when we generalize the empiric information to construct a model, we postulate that the underpinning facts are universal.
e) our model is so tentative, that, we believe, the word “speculate” fits better than "propose" or "hypothesize". We are not native English speakers, so we can be mistaken. We interpret “speculating” as adding more fantasy and unreliability than when "proposing" or "hypothesizing" something
f) Line 376: citation 109 has now been replaced for 111 in the context of "reviewed in". Thank you for careful reading.
g) In former Fig. 2 (now Fig. 1), "low ROS" meant "no stress". Deleted in the updated version. We have simplified the figure in order to remove most questions arisen. In particular, the minor fraction has been made really minor. But it should be emphasized in general, that the scheme is tentative and hypothetical.That's why we wouldn't like to detalize it, and this is our firm position. The forthcoming studies would provide us with the details in future.
h) Table 1, it was noted in the original manuscript that children are different from adults (lines 509-510). We can not explain why the sharpest difference is observed for the children group.
4) Conclusion
i) Line 569, SatII is added (SatII/III). Unfortunately, we have had no idea what can be added to this section about adaptive response.